# Cu and As(V) Adsorption and Desorption on/from Different Soils and Bio-Adsorbents

**DOI:** 10.3390/ma15145023

**Published:** 2022-07-19

**Authors:** Raquel Cela-Dablanca, Ana Barreiro, Gustavo Ferreira-Coelho, Claudia Campillo-Cora, Paula Pérez-Rodríguez, Manuel Arias-Estévez, Avelino Núñez-Delgado, Esperanza Álvarez-Rodríguez, María J. Fernández-Sanjurjo

**Affiliations:** 1Department of Soil Science and Agricultural Chemistry, Engineering Polytechnic School, University of Santiago de Compostela, 27002 Lugo, Spain; ana.barreiro.bujan@usc.es (A.B.); gustavof.coelho@gmail.com (G.F.-C.); avelino.nunez@usc.es (A.N.-D.); esperanza.alvarez@usc.es (E.Á.-R.); mf.sanjurjo@usc.es (M.J.F.-S.); 2Soil Science and Agricultural Chemistry, Faculty of Sciences, University of Vigo, 32004 Ourense, Spain; ccampillo@uvigo.es (C.C.-C.); paulaperezr@uvigo.es (P.P.-R.); mastevez@uvigo.es (M.A.-E.)

**Keywords:** bio-adsorbents, heavy metals, soil pollution, release, retention

## Abstract

This research is concerned with the adsorption and desorption of Cu and As(V) on/from different soils and by-products. Both contaminants may reach soils by the spreading of manure/slurries, wastewater, sewage sludge, or pesticides, and also due to pollution caused by mining and industrial activities. Different crop soils were sampled in A Limia (AL) and Sarria (S) (Galicia, NW Spain). Three low-cost by-products were selected to evaluate their bio-adsorbent potential: pine bark, oak ash, and mussel shell. The adsorption/desorption studies were carried out by means of batch-type experiments, adding increasing and individual concentrations of Cu and As(V). The fit of the adsorption data to the Langmuir, Freundlich, and Temkin models was assessed, with good results in some cases, but with high estimation errors in others. Cu retention was higher in soils with high organic matter and/or pH, reaching almost 100%, while the desorption was less than 15%. The As(V) adsorption percentage clearly decreased for higher As doses, especially in S soils, from 60–100% to 10–40%. The As(V) desorption was closely related to soil acidity, being higher for soils with higher pH values (S soils), in which up to 66% of the As(V) previously adsorbed can be desorbed. The three by-products showed high Cu adsorption, especially oak ash, which adsorbed all the Cu added in a rather irreversible manner. Oak ash also adsorbed a high amount of As(V) (>80%) in a rather non-reversible way, while mussel shell adsorbed between 7 and 33% of the added As(V), and pine bark adsorbed less than 12%, with both by-products reaching 35% desorption. Based on the adsorption and desorption data, oak ash performed as an excellent adsorbent for both Cu and As(V), a fact favored by its high pH and the presence of non-crystalline minerals and different oxides and carbonates. Overall, the results of this research can be relevant when designing strategies to prevent Cu and As(V) pollution affecting soils, waterbodies, and plants, and therefore have repercussions on public health and the environment.

## 1. Introduction

The increasing spreading of metals and metalloids included in the group of the so-called “heavy metals” into the soil through fertilizers, manure/slurry, sewage sludge, irrigation with wastewater, pesticides, or mining and industrial activities has given rise to concerns about their impact on the environment in general and human health in particular [1,2,3,4]. These substances enter various environmental compartments (soil, water, and air), and affect different living beings (microbial, plant, and animal communities) and may have adverse effects on individual biological receptors and populations [5]. Their toxicity is affected by the difficulties of organisms to achieve their excretion, with a tendency to bio-accumulate, and, even in cases where they do not have high concentrations in specific environments, they can reach harmful levels after passing through the food chain [6,7].

Arsenic is naturally present in certain minerals, but its presence as a pollutant in the environment can also be caused by certain human activities, such as mining, use of fossil fuels, pesticides, and herbicides. It is a semimetal or metalloid that can occur in inorganic form, with the As(III) species being the most frequent in reducing conditions, and As(V) in well-aerated media, while the organic forms (with As included in organic molecules) are quantitatively less important [8]. This element causes special concern due to its high toxicity, and it can be mobilized in the most frequent groundwater pH values, thus threatening drinking water resources [9]. It has associated chronic toxic effects, increasing the risks of developing cancers affecting the skin, lung, kidney, and liver [10,11]. Details regarding the effects of arsenic on toxicity and human health have been extensively documented in previous papers [12,13,14,15,16].

As regards Cu, it is an essential micronutrient for human beings and for plant development, but it is toxic when present at high concentrations [17,18]. It is less mobile than As, but high Cu concentrations can alter cell division in some plants, affect microbial activity, microorganism diversity, and soil ecosystem services [19,20,21,22], and cause damage to detritivore populations [23]. As regards the effects of Cu on human health, extensive reviews have been carried out in previous publications [24,25,26,27].

Soils can act as a sink for these pollutants and reduce their toxicity through adsorption, precipitation, or occlusion processes, mainly affected by soil organic matter, low crystallinity minerals, and acid–base and redox conditions [28,29].

The retention capacity of the soil can be an important factor in mitigating the toxic effects of these metals/metalloids, but, in the long term, soil adsorbent surfaces could be saturated, increasing the risk of passage to plants, water, and the food chain. To minimize this problem, different remediation strategies have been developed, mainly aimed at acting on the mobility of the pollutants [30], including the use of bio-adsorbent materials. In this regard, studies focusing on bio-adsorbents are of growing interest, as an efficient and low-cost alternative to retain the different contaminants present in soils and water. Generally, materials that are low cost and locally available in large quantities are considered a good choice to be assessed regarding their effectivity [31]. The food and agroforestry industries produce large amounts of waste and by-products, such as mussel shell, biomass combustion ash, and pine bark, which could be used for this purpose. Specifically, previous studies on pollutant retention onto biomass ash showed promising results in investigations focused on As [32,33] as well as on Cu [34,35].

In this view, the objective of this work is to study the retention of Cu and As(V) in cultivated soils with different characteristics, as well as the capacity of different by-products (oak ash, pine bark, and mussel shell) to immobilize these contaminants. The results of the research could be useful in program-appropriate practices to manage soils and low-cost by-products in order to reduce the risks of environmental contamination associated with the spreading of materials that contain both pollutants.

## 2. Materials and Methods

### 2.1. Soils and By-Products

For this research, six crop soils were selected, which were previously sampled at two areas of Galicia (NW Spain) subjected to intensive farming: S soils (sampled at Sarria, Lugo province) and AL soils (sampled at A-Limia, Ourense province). The samples were taken from the surface layer (0–20 cm), with each one being the result of combining 10 sub-samples collected in a zig-zag manner for each soil. These soils have been previously studied and described [36].

The forest by-products used in this study were oak ash from a local boiler at Lugo (Spain), pine bark (fraction less than 0.63 mm), a commercial product provided by Geolia (Madrid, Spain), and un-calcined mussel shell (<1 mm in diameter), supplied by Abonomar S.L. (Illa de Arousa, Pontevedra province, Spain). A more complete description was previously published [37].

The methods used for the characterization of soils and by-products were the following: pH in water and 0.1 M KCl (soil:solution ratio 1:2.5), using a pH-meter (pH-model 2001 Crison, Spain); C and N by elemental analysis (CHNS Truspec, Leco, St. Joseph, MI, USA); available P by the Olsen method [38]; exchangeable cations, extracted with 1 M NH_4_Cl [39] and quantified by atomic absorption/emission spectrometry; the effective cation exchange capacity (eCEC) was calculated as the sum of exchangeable Ca, Mg, Na, K, and Al; non-crystalline Al and Fe (Al_o_, Fe_o_) were extracted with ammonium oxalate acidified at pH 3. All determinations were performed in triplicate.

Table 1 and Table 2 show the main characteristics of the soils and the three by-products used, respectively.

Appendix A shows data on BET surface areas for the six soils studied, evidencing that the values were higher for S soils. Of note, although higher surface area facilitates achieving higher adsorption of a variety of substances onto soils, other factors could be of even higher relevance, as previously stated for different pollutants [40].

In addition, Appendix A shows data on BET surface areas for the three by-products, evidencing that the highest value corresponded to oak ash (1.3336 m^2^ g^−^^1^), followed by mussel shell (1.1318 m^2^ g^−1^), and being much lower for pine bark (0.3320 m^2^ g^−1^).

### 2.2. Adsorption and Desorption Experiments

To perform adsorption studies, batch-type experiments were carried out, stirring 1 g of each soil or by-product for 24 h with 40 mL of 0.005 M CaCl_2_ and with different concentrations of Cu or As(V) (100, 200, 400, 800, and 1000 μmol L^−1^), with each pollutant added individually. The solutions were prepared from analytical grade Cu(NO_3_)_2_.3H_2_O and Na_2_HAsO_4_ (Panreac, Barcelona, Spain). After 24 h of agitation, the samples were centrifuged (at 4000 rpm) and filtered. In the equilibrium solution, the dissolved organic carbon (DOC) was determined by means of UV-1201 spectroscopy (Shimadzu, Kyoto, Japan), the pH using a glass electrode (Crison, Madrid, Spain), and the concentrations of Cu or As using an ICP-MS equipment (Varian 820-NS, Palo Alto, CA, USA). The amount of Cu or As adsorbed was calculated by the difference between the added concentration and that remaining in the equilibrium solution.

Regarding desorption experiments, 40 mL of 0.005 M CaCl_2_ was added to each of the samples used in the previous adsorption tests, then stirring for 24 h, centrifuging, filtering and quantifying Cu or As(V) in the equilibrium solution, following the same methodology indicated above.

### 2.3. Data Analysis and Statistical Treatment

The experimental adsorption data were checked as regards their fitting to the Freundlich (Equation (1)), Langmuir (Equation (2)), and Temkin (Equation (3)) models:(1)$$q_{a}=K_{F}C_{eq}^{n}$$
(2)$$q_{a}=K_{F}C_{eq}^{n}$$
*q_a_* = *β* ln *K_T_* + *β* ln *C_eq_*(3)
where *q_a_* is the amount of Cu or As(V) adsorbed in equilibrium (μmol kg^−1^); *C_eq_* is the concentration of Cu or As(V) present in the solution in the equilibrium (μmol L^−1^); *K_F_* is the Freundlich affinity parameter (L^n^ µmol^1−n^ kg^−1^); *n* is the Freundlich linearity parameter (dimensionless); *K_L_* is a Langmuir parameter related to the adsorption energy (L μmol^−1^), and *q_m_* is the Langmuir’s maximum adsorption capacity (μmol kg^−1^). In addition, *β* is calculated as *RT/bt*; *bt* is the Temkin isotherm constant; *T* is Temperature (K = 298°) (25 °C); *R* is the universal gas constant (8314 Pa m^3^/mol K); and *K_T_* is the Temkin isotherm equilibrium binding constant (L g^−1^).

Desorption was expressed as the amount of Cu or As(V) desorbed (in μmol kg^−1^, and also as percentage) with respect to the amount previously adsorbed.

The statistical software R version 3.1.3 and the *nlstools* package for R [41] were used to check the fittings to the adsorption models. The SPSS 15.0 software was used to carry out bivariate Pearson correlations between adsorption and desorption data and characteristics of the sorbent materials, and multiple linear regression analyses.

## 3. Results

### 3.1. Cu and As(V) Adsorption onto Soils

The adsorption curves for the soils and bio-adsorbents studied are shown in Figure 1 and Figure 2, respectively.

Figure 1 shows a variety of shapes in the adsorption curves, with differences between the AL and S soils. In fact, these curves show that overall Cu adsorption was higher for S soils (which have higher surface area) than for AL soils, while As(V) adsorption was similar for both kinds of soils.

Figure 2 shows that Cu and As(V) adsorption results were clearly higher for oak ash as compared to pine bark and mussel shell.

Figure 3 shows the results corresponding to Cu and As(V) adsorption onto the different soils (both in absolute value and percentage) as a function of the concentration added. Considering the absolute values, it is clear that the higher the Cu or As(V) concentrations added, the higher the adsorption for all soils, while the adsorbed percentage shows a decreasing trend. Adsorption was generally higher for Cu than for As, especially in S soils.

When the highest Cu or As(V) concentrations (1600 µmol L^−1^) were added, Cu maximum adsorption values were reached in soils 51S and 71S (37,687 µmol kg^−1^ and 44,019 µmol kg^−1^, respectively), while for As(V), the highest scores corresponded to soils 50AL and 71S (17,076 µmol kg^−1^ and 22,980 µmol kg^−1^, respectively) (Figure 3). In contrast, the minimum Cu adsorption corresponded to soils 19AL and 3AL (5963 µmol kg^−1^ and 12,523 µmol kg^−1^, respectively), while for As(V), the minima were for soils 19AL and 6S (6290 µmol kg^−1^ and 5868 µmol kg^−1^, respectively).

Regarding percentage adsorption, within AL soils, the one with the highest organic matter content (soil 50AL, Table 1) adsorbed about 90% of Cu for the three lowest doses added, while this percentage dropped to 57% for the highest dose; however, for soil 19AL (the one with the lowest organic matter content), Cu adsorption never exceeded 56%, being less than 12% for the highest dose. The progressive decrease in the adsorption rate affecting these three AL soils could be related to a saturation of the adsorption sites, many of which would be functional groups in organic compounds, and that decrease would be more pronounced for those soils with a lower organic matter content. In S soils, the adsorption was close to 100% for the three lowest doses of Cu added, decreasing to 82% in the soil with the highest organic matter content (soil 71S) and to 56% in soil 6S when the maximum Cu dose was added, again due to the saturation of the functional groups involved in adsorption, many of which lie in organic matter.

Table 3 shows data corresponding to Cu and As(V) adsorption for the various initial concentrations added of both pollutants to the soils studied, in parallel to data corresponding to pH and DOC values in the equilibrium solution. 

### 3.2. Cu and As(V) Desorption from Soils

Figure 4 shows the amounts of Cu and As(V) desorbed from the soils as a function of the concentrations added. As the added dose of each element increased, both the amount and the percentage desorbed were higher. All AL soils had a similar desorption of both elements, while S-zone soils (with higher pH) desorbed much more As than Cu (Figure 4).

In relation to Cu, desorption was much higher from AL soils than from S soils (the latter having a higher surface area). The maximum percentage values for AL soils were between 39% of the soil with less organic matter (19AL) and 12% of the one containing most organic matter (soil 50AL), while the range for the S zone was narrower: between 15% (soil 6S) and 5% (soils 51S and 71S). In general, soils with low desorption values match those with high adsorption scores.

### 3.3. Cu and As(V) Adsorption onto the Three By-Products

Figure 5 shows Cu and As(V) adsorption onto the three by-products as a function of the concentration added. Adsorption was always much higher for Cu than for As(V), especially for pine bark and mussel shell, while for oak ash, the differences were clearly smaller, although becoming more evident as the added dose increased.

In relation to Cu, its adsorption increased in all cases as a function of the concentration of Cu added. At high doses of the pollutants, the differences among the by-products were more apparent, with the highest adsorption corresponding to oak ash, followed by mussel shell and pine bark. Mussel shell and especially oak ash have a pH clearly higher than that of pine bark (Table 2), which may influence the different adsorption on the three by-products. As the pH increases, the negative charge in the variable charge colloids rises, favoring cationic retention, as mentioned above for soils. In this sense, non-crystalline minerals, which provide a high variable charge, were much more abundant in oak ash than in mussel shell, which could explain why oak ash adsorbed more Cu.

### 3.4. Cu and As(V) Desorption from the Three By-Products

Figure 6 shows data corresponding to Cu and As(V) desorption from oak ash, pine bark, and mussel shell. For Cu, the desorption sequence was: pine bark > mussel shell ≥ oak ash (inverse to that of adsorption). Cu desorption was low from oak ash and mussel shell, with the maximum desorption value being 30.5 µmol kg^−1^ for oak ash and 70.16 µmol kg^−1^ for mussel shell. For pine bark, desorption rose remarkably when increasing the concentration of added Cu, with the minimum value being 246.66 µmol kg^−1^ and the maximum reaching 6188.55 µmol kg^−1^. This by-product has a very high concentration of organic C, clearly higher than those of oak ash and mussel shell, with organic matter being responsible for retaining much of the Cu added.

### 3.5. Fitting of Cu and As(V) Experimental Data to Different Adsorption Models

Cu and As(V) adsorption can be partially fitted to the Langmuir (Table 4), Freundlich (Table 5), and Temkin (Table 6) models. The three models for both elements have R^2^ values ranging between 0.784 and 0.999 for Langmuir, 0.845–0.999 for Freundlich, and 0.732–0.999 for Temkin.

## 4. Discussion

### 4.1. Cu and As(V) Adsorption onto Soils

The influence of organic groups on Cu adsorption shown in the current research has already been pointed out by other authors [42], and, in the current study, it is supported by the significant and positive correlation obtained between Cu adsorption (when the maximum dose is added) and soil organic matter content (r = 0.53, *p* < 0.05). However, there are other factors that influence adsorption, as indicated by the fact that both the amount and the percentage of Cu adsorbed were always higher in S soils (which are those that have higher pH values, as well as a higher surface area) compared to AL soils. This would indicate that adsorption is a pH-dependent process and is related to the influence of this parameter on the solubility of metal ions and also on the ionization state of functional groups of adsorbent surfaces (variable charge components) [43,44]. In this sense, soils with a higher pH have more negative charges in the variable charge components, mainly in organic matter, but also in non-crystalline components, which are also much more abundant in S soils (Table 1).

The effect of non-crystalline Fe and Al minerals on Cu adsorption has been reported by several authors [45,46] and attributed to Cu-specific complexation and adsorption reactions onto non-crystalline oxy-hydroxides. Supporting the latter, in the present study, a significant (*p* < 0.05) and positive correlation was obtained between Cu adsorption and Fe_o_ (r = 0.812), and also with the sum of Fe_o_ and Al_o_ (r = 0.819), parameters that estimate the content of non-crystalline minerals. A significant and positive correlation was also found between Cu adsorption and eCEC (*p* < 0.01, r = 0.946). In addition, performing a successive steps regression, it was obtained that eCEC (in which organic matter and non-crystalline minerals are of great importance) explains 87% of Cu adsorption. This clear influence of eCEC is indicative of the importance of charges present in soil colloids for adsorption. It is also worth noting the decrease in pH in the equilibrium solution as the adsorption of Cu increases (Table 3), finding a significant and negative correlation between both parameters, with *p* < 0.01 for soils 51S (r = −0.99), 71S (r = −0.99), and 19AL (r = −0.96), and with *p* < 0.05 for soils 3AL (r = −0.94), 50AL (r = −0.96), and 6S (r = −0.897), which could be related to the proton exchange taking place in the Cu adsorption process.

Regarding As(V), differences in adsorption were no so clear for S and AL soils (Figure 3). This would indicate that soil pH does not have such an obvious effect on As adsorption, compared to Cu. Several authors indicate that As(V) can be adsorbed over a wide range of pH. Specifically, Stanić et al. [47], using zeolite as adsorbent, reported a range between 4.0 and 11.0, and Mamindy-Pajany et al. [48] found 100% adsorption for As on hematite in a pH range between 2 and 11. However, other authors have reported ranges not as wide, such as 6–8 for alumina impregnated with La^3+^ and Y^3+^, or a range of 2–4 for molybdenum-impregnated chitosan [49,50]. Recently, Yusof et al. [51] reported a range of 3–7 using palm oil combustion ash. It should be noted that, under oxidizing conditions and at a low pH, arsenate is dominant, mainly as H_2_AsO_4_^−^, while as the pH increases the predominant species would be HAsO_4_^2−^ [52,53,54]. In acid soils, such as those in the AL zone of the current study, the H_2_AsO_4_^−^ species would be adsorbed on variable-charge colloids, which would be positively charged due to protonation taking place at that low pH prevailing, and then adsorption could take place by means of electrostatic attraction. Within these colloids would be non-crystalline minerals [53], which are more abundant in soils 3AL and 50AL compared to 19AL, coinciding with the highest adsorption taking place in the former (Table 1 and Figure 3). Several authors have found that arsenates are strongly adsorbed to these kinds of Al (and especially Fe) compounds, and the adsorbed amounts can be significant even with low concentrations of As present in the liquid phase [55,56]. In the current study, the zone S soils have high concentrations of non-crystalline minerals of Fe and Al compounds (mainly oxy-hydroxide), and, although these soils have higher pH values than those of zone AL, it is below the pH of the zero point of charge (zpc) of these minerals (between 8.7 and 9.1) [57], with which these colloids would be positively charged, and adsorption could also occur by electrostatic attraction. Other colloids would present a negative charge at pH around 6, as would happen with soil organic matter. In these cases, adsorption can be performed through a cationic bridge and/or by ligand exchange. In relation to the latter, significant and positive correlations were found between the pH in the equilibrium solution and As adsorption in soils 3AL, 50AL, and 51S (r = 0.987, 0.992, and 0.997, respectively, *p* < 0.01) and in soil 71S (r = 0.978, *p* < 0.05) (Table 3). This increase in the equilibrium pH in the adsorption process is in line with findings previously reported [53,58], which would be justified by an exchange of ligands between the species H_2_AsO_4_^−^ or HAsO_4_^2−^ and OH^−^ groups, which are released in the solution.

In addition, the influence of organic matter on the adsorption of As is also present. In the S zone, adsorption was higher in the soil with the highest organic matter content (soil 71S), compared with the other two (soils 51S and 6S). However, this influence is not as obvious as for Cu, since both 50AL and 3AL soils (with very different concentrations of organic C, Table 1) have a similar As adsorption capacity (Figure 1). As mentioned above, expressing As(V) adsorption as a percentage (Figure 3), a decrease is observed when the concentration added rises, which is due to the afore-mentioned saturation of the adsorbent surfaces. In AL soils, the As(V) adsorption percentage dropped from 75–80% to 30% (for soils 3AL and 50AL), and from 30–40% to 11% for the one with less organic matter content (19AL). In S soils, the largest decrease in the As(V) adsorption percentage, due to the rise in the concentration added, took place for soil 6S (from 100% to 10%).

### 4.2. Cu and As(V) Desorption from Soils

It should be borne in mind that different soil factors can influence the desorption of metals and metalloids. Several authors highlight the influence of pH on the desorption of heavy metals from soils [59,60], since, with increasing pH, Cu desorption would decrease linearly, which in the present study could explain, in part, the differences observed between the soils of the two zones. Liang et al. [61], studying Cu desorption in rice-growing soils, attributed the decrease in desorption to the rise in soil eCEC, and this influence of eCEC was again pointed out by Zhanget al. [62]. This coincides with that obtained for the soils of the present study, since those with higher eCEC (S Soils) (Table 1) are those that desorbed less Cu (Figure 4).

Regarding As(V) (Figure 4), the high desorption taking place for all S soils (which are in the range of 3800–8400 µmol kg^−1^ for the maximum dose added) contrasts with the lower desorption found for AL soils (between 424 and 676 µmol kg^−1^). This represents a percentage of desorption that did not exceed 40% in the AL zone, while in the S zone, the desorbed proportion reached maxima between 40 and 67%. Again, the soils showing the highest As desorption are those that adsorbed the least (Figure 3 and Figure 4). By conducting a bivariate correlation study to find out how different soil parameters influence As(V) desorption, a significant (*p* < 0.05) and positive correlation was obtained with soil pH (r = 0.837) and non-crystalline Fe and Al compounds (r = 0.848). There is also a significant (*p* < 0.05), but negative (r = −0.875), correlation with the available phosphorus content. These results support the importance of pH in the desorption processes. As the pH increases, the positive charges on the colloidal surfaces (including those of the non-crystalline Fe and Al compounds) become negative, hindering the adsorption of As(V) in anionic form, causing the bonds to be more labile, while the opposite happens when acidity increases. This would explain the greater desorption of As(V) from S soils, with higher pH. A negative correlation with available P could indicate that arsenate ions compete with phosphate ions (which are often found bound to non-crystalline Fe and Al compounds with different adsorption energies), resulting in an increase in available P and an adsorption of As(V) with different holding forces. Results similar to these were found by Rahman et al. [63] studying the adsorption and release of As in contaminated soils, observing higher As(V) release as the pH increased, attributing it to electrostatic repulsion. These authors also pointed to the strong competition of arsenate with phosphate, which is much higher than with sulfate.

### 4.3. Cu and As(V) Adsorption onto the Three By-Products

According to Boim et al. [64], the decrease in Cu mobility as pH increases is due to the formation of insoluble complexes, and they also highlight the importance of non-crystalline Fe and Al oxy-hydroxides in the adsorption of Cu^2+^. In a study carried out on vineyard soils amended with mussel shell [65], a decrease in the available Cu was observed, which was related to the increase in soil pH, although it could also be affected by the direct adsorption of Cu onto the added mussel shell [66]. Pine bark, the most acidic material among the three by-products, has abundant organic matter with different functional groups, some of which may have a negative charge even at pH values < 3 [67], which would explain the adsorption values of Cu being just slightly lower than those corresponding to mussel shell. These reactive functional groups present in pine bark are progressively saturated as the added Cu dose rises, as indicated by the decrease in the percentage of Cu adsorbed (decreasing from 87% to 68%) (Figure 5).

In contrast, oak ash and mussel shell adsorbed 100% of the amounts of contaminants added, except for the highest dose, where adsorption decreased to 79% for mussel shell. Furthermore, Table 7 shows that pH in the equilibrium solution decreased with the increasing concentration of adsorbed Cu, as mentioned for soils, obtaining a significant and negative correlation between both parameters, with r = −0.964 (*p* < 0.01) for oak ash and r = −0.840 (*p* < 0.05) for mussel shell. Šoštarić et al. [68] also found a decrease in pH after the adsorption of different metals onto apricot peels, which was caused by the release of H^+^ due to strong competition with cationic metals, suggesting the intervention of ion exchange processes.

Table 7 also shows that, regarding the values of dissolved organic carbon (DOC), they tend to decrease for oak ash and pine bark when Cu adsorption rises. In fact, for pine bark, a significant negative correlation is obtained between DOC and adsorption (r = −0.844 and *p* < 0.05). This could be related to the high affinity between Cu and organic matter, forming organometallic complexes, which could move to the solid phase [69], this being another mechanism for Cu retention.

Regarding As(V), Figure 5 shows that oak ash is also the material with the highest adsorption, which increases as the concentration added rises. As(V) adsorption is much lower on mussel shell, and especially on pine bark. Given the high pH values corresponding to oak ash and mussel shell (Table 2), the predominant As species will be HAsO_4_^2−^ [52], and non-crystalline components will be negatively charged, meaning that the bond between the anionic As and these surfaces could take place by means of a cationic bridge. Oak ash contains, in addition to carbonates, oxides of Ca, Fe, and other elements, and these oxides would contribute to As(V) adsorption either by physical mechanisms or by chemical reactions [63]. As discussed above, the presence of a high concentration of oxalate-extractable Al and Fe (non-crystalline Fe and Al compounds) (Table 2) could also explain the high adsorption taking place on oak ash.

A variety of authors have found that arsenates are strongly adsorbed to these compounds (especially to non-crystalline Fe), and the amounts adsorbed can be relevant, even when low As concentrations are present in the liquid phase [55,56]. Furthermore, for oak ash and pine bark, the pH of the equilibrium solution shows a tendency to decrease as As(V) adsorption rises (Table 7), while pH increases in the case of mussel shell, with no significant correlation for any of the three by-products. Several studies have reported an increase in the equilibrium pH for As(V) adsorption due to exchange with OH^−^ groups, as explained above. The fact that in some cases this does not occur would indicate that other anions (SO_4_^2−^, PO_4_^3−^, or organic anions) are released, or that other mechanisms are involved in As adsorption, such as adsorption on calcite and Van der Waals forces, where OH^−^ groups are not released [70].

The good results regarding the retention of both pollutants in oak ash could also be reltated to its higher BET surface area (Appendix A), as previously pointed out for tetracycline antibiotics using the same three bio-adsorbents [71].

Other authors carried out studies using low-cost sorbents including ash, but some of them were clearly different materials compared to the oak ash used in the current work. As an example, Tsang et al. [72] found good results for coal fly ash as regards As stabilisation, although not successful for Cu retention. Mitchell et al. [73] found a reduction in soluble As and Cu by means of cementitious aggregation of wood ashes, although the authors indicate that the extent is metal(loid)-specific when amended to soils. Park et al. [74] studied fly and bottom ash from wood pellet thermal power plants, finding that these by-products have a high potential for heavy metal removal, although the authors focused specifically on Cd. In addition, the quality of the bio-adsorbents is relevant, as shown by Lucchini et al. [75] working with ash derived from Cu-based preservative-treated wood, where the authors reported that these by-products can lead to extremely high Cu concentrations in soil and negatively affect plant growth.

In view of the layout of some of the isotherm graphs, it could be considered that the concentrations of some of the sorbents and/or the concentration range of the sorbates were not optimal, influencing the accuracy of the fitting of various models. In this regard, we have published previous papers dealing with these and other aspects related to Cu and As adsorption/desorption studies, using similar values to those of the current work for molar concentrations of these pollutants, which facilitates the easier comparison of retention efficacy, although other concentrations and ranges would be cleary interesting for future investigation, to shed further and more specific light on the overall processes. As examples, the references [76,77,78] correspond to some of these papers.

### 4.4. Cu and As(V) Desorption from the Three By-Products

According to [79], Cu binds to OC occupying high-affinity sites when Cu activity is low, but if that activity increases, Cu would also occupy low-affinity sites, which would facilitate desorption, and this could explain the obvious increase in desorption taking place only in pine bark as the concentration of Cu added rises. In relation to the percentage desorbed (Figure 6), a gradual increase is observed for pine bark as the dose of Cu added rises, reaching a maximum value of 15%. For oak ash and mussel shell, desorption rates were very low, not exceeding 2% in any case.

Regarding As(V), the desorption sequence was: mussel shell ≥ pine bark> oak ash. For mussel shell, As(V) desorption increases as the added concentration rises (Figure 6), ranging from 186.25 to 1556.26 µmol kg^−1^, which corresponds to a value of 34% as maximum desorption. For pine bark, desorption reaches 35%, while for oak ash As(V) desorption was practically zero, probably due to the strong adsorption taking place at its high pH values, especially adsorption on the Fe and Al oxy-hydroxides very abundant in this by-product (Table 2), as indicated above. This shows the excellent adsorption capacity of oak ash for both Cu and As(V).

### 4.5. Fitting of Cu and As(V) Experimental Data to Different Adsorption Models

Table 4 shows that the Langmuir *q_m_* parameter, related to adsorption capacity, is generally higher for Cu than for As(V), and for soils and by-products having higher pH values. In addition, among the most acidic soils and sorbents, *q_m_* is higher for those with higher organic matter content, which will be the ones requiring higher concentrations of these elements to become saturated [80]. The *q_m_* values obtained for Cu are significantly correlated with the N content (r = 0.838, *p* < 0.01), N being related to organic matter, which corroborates the role of organic substances in Cu adsorption. Regarding As(V), *q_m_* values are significantly and positively correlated with the cation exchange capacity (r = 0.905), Ca (r = 0.864), K (r = 0.913), and pH_KCl_ (r = 0.71), which could suggest that the As adsorbed is in anionic form, with adsorption taking place through a cationic bridge on the negatively charged components of variable charge (mainly organic matter and non-crystalline minerals). 

As regards the Langmuir *K_L_* parameter, its value was higher for Cu than for As(V), which suggests a higher adsorption energy for Cu [81]. *K_L_* is significantly and positively correlated with pH in water (r = 0.95, *p* < 0.01, for Cu, and r= 0.74, *p* < 0.05, for As), indicating an increase in the retention energy with increasing negative soil charge.

The values of the Freundlich parameter *K_F_* parameter, related to the multilayer adsorption capacity of a given adsorbent [82], were also much higher for Cu than for As(V) (Table 5). In general, the highest *K_F_* values were found in those soils and by-products having higher pH, with a significant correlation (*p* < 0.01) between *K_F_* and pH_H2O_ (r = 0.941 and 0.85 for Cu and As, respectively) and also between *K_F_* and a parameter closely related to soil pH, especially in soils with variable charge, which is eCEC (r = 0.89 and 0.88 for Cu and As, respectively). The *n* Freundlich parameter indicates the reactivity and heterogeneity of the active sites of the adsorbent, with Table 5 showing that the *n* value is always lower than 1 (specifically, it varies between 0.219 and 0.649), except for Cu adsorption on oak ash. This indicates the existence of non-linear and concave adsorption curves, with heterogeneous adsorption surfaces, which leads to a decrease in adsorption sites as the added metal/metalloid concentration increases [83], which coincides with the decrease in the adsorbed percentage as the dose of Cu and As added increases.

Temkin’s model is related to the adsorption energy and is characterized by a uniform distribution of binding energies up to a maximum level [84]. It is also assumed that this energy decreases linearly with surface coverage due to adsorbent–adsorbate interactions. The Temkin parameters in the current study show R^2^ values generally >0.80. According to [85], values of the Temkin constant (*bt*) lower than 20 KJ mol^−1^ would indicate the existence of physical adsorption processes. Table 4 shows that, in general, *bt* values are always higher, which would justify the presence of chemisorption reactions [84]. According to [86], these high *bt* values would indicate a high degree of interaction between both pollutants (As and Cu) and the adsorbents used in the present study.

## 5. Conclusions

In the soils and by-products used in this study, adsorption was higher for Cu than for As(V). In the soils, pH and organic matter content were the most influential factors as regards Cu adsorption, which increased with both parameters. In addition, Cu in cationic form would be adsorbed by binding to the negatively charged sites generated by raising the pH in the variable charge components (mainly organic matter and low-crystallinity minerals). Regarding As(V) adsorption, the influence of pH and organic matter was not so clear, since, at a low pH, As in anionic form binds to the positively charged colloids, while at higher pHs, other mechanisms intervene, such as cationic bridges or ligand exchange. For As(V) and, to a lesser extent, for Cu, the percentage of adsorption decreases with increasing dose of the added pollutant, indicating the saturation of the adsorption sites. Regarding desorption, in soils with more acidic pH (AL soils), Cu and As desorption were similar, while in soils with higher pH (S soils), more As is desorbed. Overall, oak ash performed as an excellent Cu and As(V) adsorbent and could be used in soil and water decontamination processes, possibly due to its high pH and content of carbonates, oxides, and non-crystalline minerals. Mussel shell and pine bark could also be used to retain Cu, but its capacity to adsorb As was low and its desorption was high. The adsorption data for both elements can be partially fitted to the Langmuir, Freundlich, and Temkin models. The values of the different parameters of the equations indicate a higher adsorption energy for Cu onto the sorbent surfaces, compared to As(V), and the existence of heterogeneous adsorbent surfaces with the gradual saturation of the adsorption sites, as well as the predominance of chemisorption reactions. In addition, the high correlations obtained among the different parameters of the equations and parameters of the sorbents support the influence of pH, exchange cations, as well as organic matter and non-crystalline Fe and Al oxy-hydroxides in Cu and As(V) adsorption. These results can be considered relevant to program an appropriate management of soils affected by Cu and As(V) pollution, as well as the use of low-cost bio-adsorbents, such as those tested in this study. In future research, soils with different characteristics could be evaluated, as well as other bio-adsorbents and/or study conditions, and on the other hand, complementary studies could be designed in order to advance the elucidation of the mechanisms that intervene in the adsorption processes of both contaminants in the sorbent materials under consideration.

## Figures and Tables

**Figure 1 materials-15-05023-f001:**
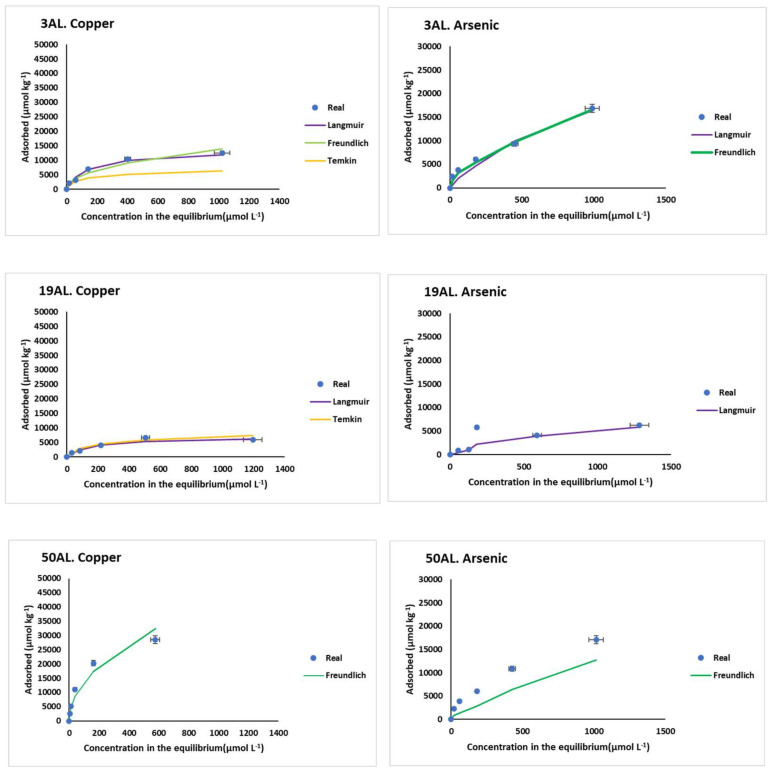
Cu and As(V) adsorption curves and selected graphical fittings to the various adsorption models for the six soils studied. Error bars represent twice the standard deviation of the mean (*n* = 3). When bars are not visible, they are smaller than the symbols.

**Figure 2 materials-15-05023-f002:**
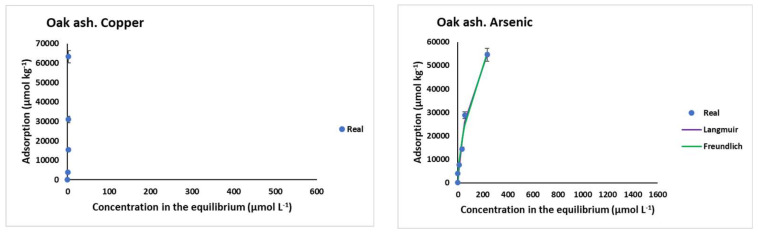
Cu and As adsorption curves and selected graphical fittings to the various adsorption models for the three bio-adsorbents studied. Error bars represent twice the standard deviation of the mean (*n* = 3). When bars are not visible, they are smaller than the symbols.

**Figure 3 materials-15-05023-f003:**
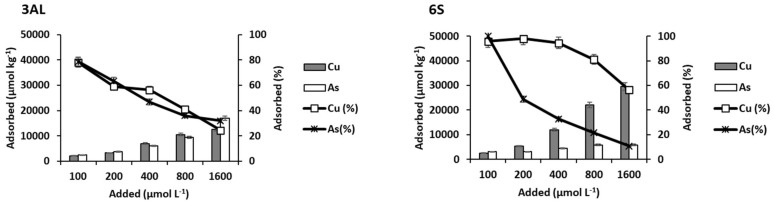
Cu and As (V) adsorption, expressed in µmol kg^−1^ and as percentage, for the soils studied, as a function of the pollutant concentrations added. Error bars represent twice the standard deviation of the mean (*n* = 3). When bars are not visible, they are smaller than the symbols.

**Figure 4 materials-15-05023-f004:**
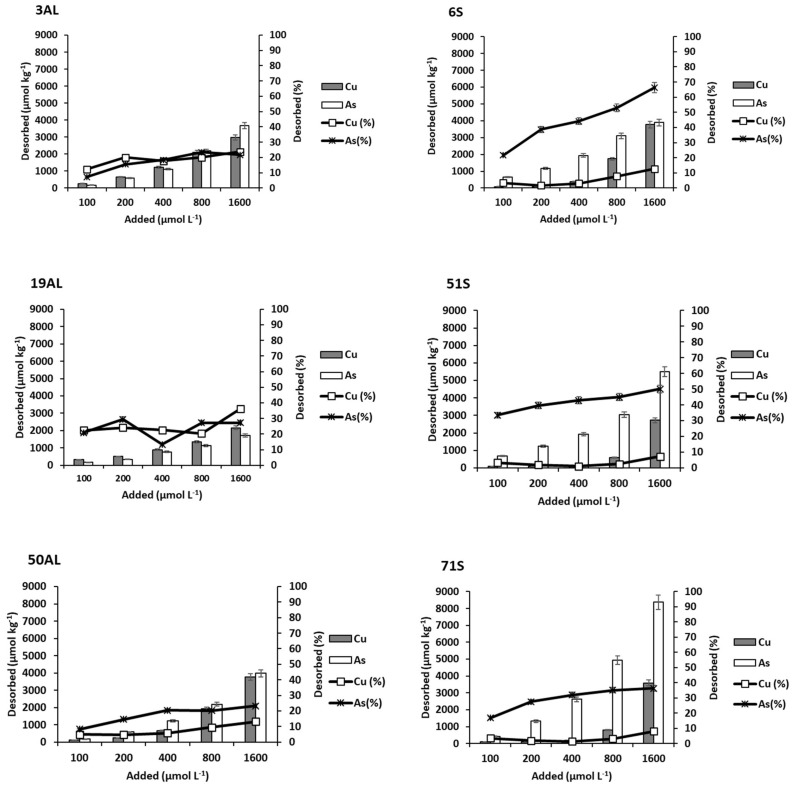
Cu and As (V) desorption, expressed in µmol kg^−1^ and as percentage, for the soils studied, as a function of the pollutant concentrations added. Error bars represent twice the standard deviation of the mean (*n* = 3). When bars are not visible, they are smaller than the symbols.

**Figure 5 materials-15-05023-f005:**
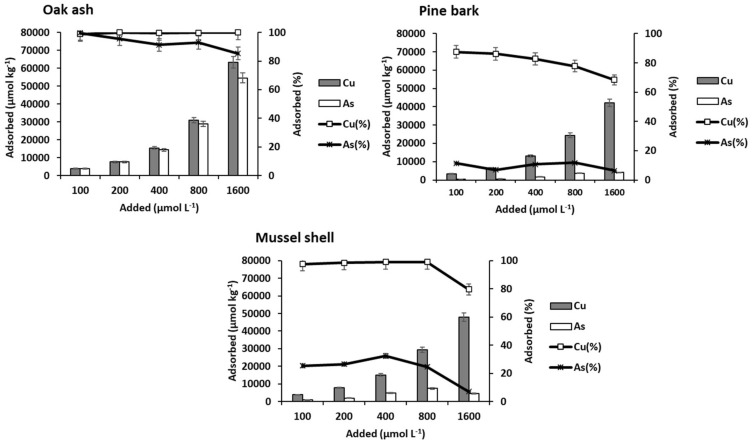
Cu and As (V) adsorption, expressed in µmol kg^−1^ and as percentage, for the three bio-adsorbents studied, as a function of the pollutant concentrations added. Error bars represent twice the standard deviation of the mean (*n* = 3). When bars are not visible, they are smaller than the symbols.

**Figure 6 materials-15-05023-f006:**
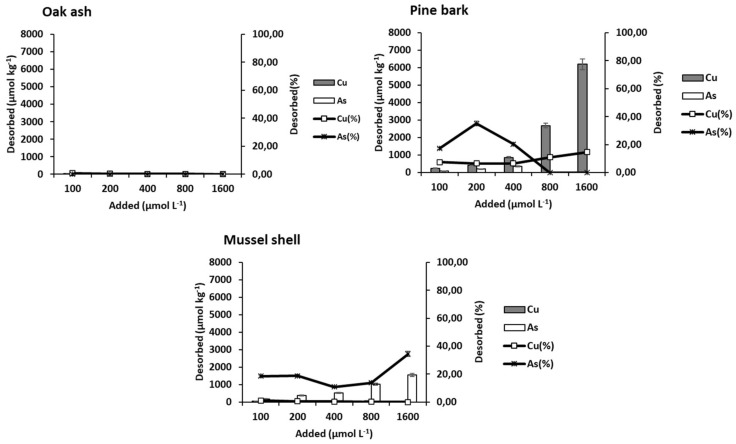
Cu and As (V) desorption, expressed in µmol kg^−1^ and as percentage, for the three bio-adsorbents studied, as a function of the pollutant concentrations added. Error bars represent twice the standard deviation of the mean (*n* = 3). When bars are not visible, they are smaller than the symbols.

**Table 1 materials-15-05023-t001:** Main characteristics of the six soils studied. Average values (*n* = 3) with coefficients of variation always lower than 5%.

Parameter	Units	Soil					
		3AL	19AL	50AL	6S	51S	71S
pH_H2O_		4.74	4.80	4.49	6.33	7.06	6.24
pH_KCl_		4.30	4.25	4.00	5.86	6.39	5.44
Ca_e_	cmol_c_ kg^−1^	2.24	1.53	5.94	12.86	9.89	12.79
Mg_e_	cmol_c_ kg^−1^	0.64	0.41	1.48	1.13	0.97	2.88
Na_e_	cmol_c_ kg^−1^	0.35	0.25	0.42	0.36	0.28	0.41
K_e_	cmol_c_ kg^−1^	1.00	1.27	1.14	0.61	1.40	1.20
Al_e_	cmol_c_ kg^−1^	1.68	0.61	2.66	0.00	0.01	0.11
eCEC	cmol_c_ kg^−1^	5.92	4.08	11.64	14.96	12.54	17.38
Al saturation	%	28.43	15.00	22.83	0.00	0.05	0.06
P	mg kg^−1^	117.90	225.43	135.90	71.42	120.03	96.77
N	%	0.31	0.09	0.84	0.23	0.19	0.48
C	%	3.39	1.07	10.92	1.98	1.75	6.88
OM	%	5.84	1.84	18.83	3.41	3.02	11.86
C/N		10.94	11.89	13.00	8.44	9.05	14.21
Sand	%	54.72	64.72	58.72	29.28	27.28	61.28
Silt	%	26.00	14.00	16.00	49.28	51.28	23.28
Clay	%	19.28	21.28	25.28	21.44	21.44	15.44
Al_o_	mg kg^−1^	5040.0	855.0	2995.0	18,377.5	15,755.7	50,593.5
Fe_o_	mg kg^−1^	2585.0	1150.0	1430.0	56,423.8	42377.4	73,095.9

Ca_e_, Mg_e_, Na_e_, K_e_, and Al_e_ = exchangeable concentrations of the elements; Al_o_ and Fe_o_ = Al and Fe concentration after extraction with ammonium oxalate.

**Table 2 materials-15-05023-t002:** Main characteristics of the three by-products used. Average values (*n* = 3) with coefficients of variation always lower than 5%.

Parameter	Unit	Oak Ash	Pine Bark	Mussel Shell
C	%	13.23	48.70	11.43
N	%	0.22	0.08	0.21
C/N		60.13	608.75	55.65
pH_H2O_		11.31	3.99	9.39
pH_KCl_		13.48	3.42	9.04
Ca_e_	cmol_c_ kg^−1^	95.00	5.38	24.75
Mg_e_	cmol_c_ kg^−1^	3.26	2.70	0.72
Na_e_	cmol_c_ kg^−1^	12.17	0.46	4.37
K_e_	cmol_c_ kg^−1^	250.65	4.60	0.38
Al_e_	cmol_c_ kg^−1^	0.07	1.78	0.03
Al saturation	%	0.02	11.91	0.11
eCEC	cmol_c_ kg^−1^	361.17	14.92	30.25
P-Olsen	mg kg^−1^	462.83	70.45	54.17
Al_o_	mg kg^−1^	8323.00	315.00	178.33
Fe_o_	mg kg^−1^	4233.00	74.00	171.00

Ca_e_, Mg_e_, Na_e_, K_e_, and Al_e_ = exchangeable concentrations of the elements; Al_o_ and Fe_o_ = Al and Fe concentration after extraction with ammonium oxalate.

**Table 3 materials-15-05023-t003:** Values of Cu and As(V) adsorption (Q) as well of pH and DOC in the equilibrium solution for the various Cu and As(V) initial concentrations (C_0_) added to the soils.

		Cu			As(V)		
Soil	C_0_ µmol L^−1^	Q µmol kg^−1^	pH	DOC mg L^−1^	Q µmol kg^−1^	pH	DOC mg L^−1^
3AL	0.00	0.00	4.75	0.08	0.00	4.76	0.19
	100	2127.04	4.64	0.20	2394.39	4.78	0.19
	200	3225.97	4.46	0.13	3764.03	4.97	0.23
	400	6951.04	4.43	0.13	6037.21	5.17	0.16
	800	10,467.99	4.25	0.12	9344.48	5.37	0.17
	1600	12,523.80	4.10	0.10	16,882.19	5.92	0.16
19AL	0.00	0.00	4.71	0.20	0.00	5.00	0.08
	100	1433.70	4.57	0.13	888.39	5.19	0.14
	200	2088.81	4.45	0.19	1153.99	5.33	0.21
	400	3969.43	4.34	0.29	5778.18	5.74	0.09
	800	6659.33	4.27	0.17	4126.55	6.11	0.12
	1600	5963.30	4.07	0.28	6290.39	6.60	0.09
50AL	0.00	0.00	4.27	0.26	0.00	4.50	0.13
	100	2524.26	4.24	0.19	2272.59	4.37	0.18
	200	5106.61	4.13	0.26	3939.78	4.38	0.20
	400	11,086.25	4.03	0.28	6058.32	4.45	0.28
	800	20,244.49	3.79	0.34	10,856.87	4.66	0.23
	1600	28,481.45	3.69	0.23	17,075.78	4.89	0.22
6S	0.00	0.00	5.61	0.20	0.00	5.88	0.19
	100	2558.33	5.66	0.19	3048.25	5.87	0.12
	200	5253.62	6.26	0.13	3015.26	6.33	0.08
	400	12,030.50	5.57	0.15	4393.79	6.44	0.09
	800	22,000.77	5.07	0.13	5849.58	6.73	0.09
	1600	29,540.60	4.75	0.17	5866.52	6.88	0.06
51S	0.00	0.00	6.04	0.16	0.00	6.61	0.15
	100	2615.99	6.09	0.18	2016.28	6.80	0.14
	200	5202.90	6.06	0.19	3091.30	6.83	0.12
	400	12,205.87	5.93	0.14	4485.69	6.85	0.09
	800	24,061.91	5.55	0.13	6775.33	6.96	0.12
	1600	37,687.91	5.09	0.11	10,962.17	7.18	0.08
71S	0.00	0.00	5.49	0.29	0.00	6.00	0.11
	100	2618.82	5.53	0.20	2516.39	5.85	0.09
	200	5282.88	5.49	0.19	4786.28	5.84	0.13
	400	11,962.35	5.36	0.16	8174.64	5.84	0.14
	800	25,655.79	5.13	0.18	14,047.81	5.92	0.13
	1600	44,019.57	4.69	0.15	22,979.85	6.07	0.13

**Table 4 materials-15-05023-t004:** Fitting to the Langmuir model of experimental data corresponding to Cu and As(V) adsorption onto the soils and bio-adsorbents used.

Soil/Bio-Adsorbent		Langmuir Parameter				
		*q_m_* (µmol kg^−1^)	Error-1	*K_L_* (L µmol^−1^)	Error-2	R^2^
3AL	Cu	13,400.09	3292.03	0.0075	0.003	0.935
	As(V)	27,759.87	9142.35	0.0014	0.0009	0.967
19AL	Cu	7020.96	1227.60	0.0065	0.003	0.958
	As(V)	9102.93	6017.42	0.0015	0.006	0.837
50AL	Cu	27,463.77	2525.40	0.021	0.003	0.992
	As(V)	25,379.96	4566.65	0.002	0.001	0.985
6S	Cu	28,242.77	6241.05	0.041	0.015	0.945
	As(V)	6457.63	295.59	0.010	0.002	0.992
51S	Cu	44,100.55	11,594.77	0.028	0.010	0.950
	As(V)	14,417.56	3161.04	0.002	0.001	0.962
71S	Cu	49,174.35	2524.46	0.033	0.005	0.996
	As(V)	32,341.86	6030.63	0.003	0.001	0.982
Oak ash	Cu	-	-	-	-	-
	As(V)	83,740.12	13,748.45	0.008	0.003	0.987
Pine bark	Cu	65,560.70	3016.86	0.0035	0.0003	0.999
	As(V)	7298.80	2723.54	0.0010	0.0007	0.958
Mussel shell	Cu	50,467.87	6290.40	0.1021	0.0364	0.959
	As(V)	6925.85	2393.08	-	-	0.784

*q_m_*: maximum adsorption capacity; *K_L_*: constant related to the intensity of interaction adsorbent/adsorbate; R^2^: coefficient of determination; -: error too high for fitting.

**Table 5 materials-15-05023-t005:** Fitting to the Freundlich model of experimental data corresponding to Cu and As(V) adsorption onto the soils and bio-adsorbents used.

Soil/Bio-Adsorbent		Freundlich Parameter				
		*K_F_* (L^n^ µmol^1−n^ kg^−1^)	Error-1	*n*	Error-2	R^2^
3AL	Cu	605.97	161.97	0.453	0.051	0.970
	As(V)	317.97	123.29	0.571	0.059	0.991
19AL	Cu	-	-	0.354	0.114	0.902
	As(V)	-	-	-	-	-
50AL	Cu	1420.25	290.32	0.492	0.047	0.969
	As(V)	396.35	71.49	0.546	0.028	0.997
6S	Cu	4334.21	1250.35	0.308	0.051	0.975
	As(V)	1302.36	548.87	0.219	0.066	0.938
51S	Cu	2042.58	750.21	0.527	0.103	0.845
	As(V)	332.74	77.00	0.491	0.035	0.996
71S	Cu	4695.40	1535.83	0.413	0.065	0.977
	As(V)	679.50	74.58	0.522	0.017	0.999
Oak ash	Cu	-	-	-	-	-
	As(V)	2481.83	963.55	0.568	0.076	0.986
Pine bark	Cu	1013.92	204.18	0.601	0.0342	0.997
	As(V)	-	-	0.649	0.1843	0.933
Mussel shell	Cu	9586.92	4114.71	0.282	0.084	0.899
	As(V)	-	-	-	-	-

*K_F_*: parameter related to the adsorption capacity; *n*: parameter related to adsorbent heterogeneity; R^2^: coefficient of determination; -: error too high for fitting.

**Table 6 materials-15-05023-t006:** Fitting to the Temkin model of experimental data corresponding to Cu and As(V) adsorption onto the soils and bio-adsorbents used.

Soil/Bio-Adsorbent		Temkin Parameters				
		*bt*	Error-1	*K_t_* (L/g)	Error-2	R^2^
3AL	Cu	1249.03	329.44	0.16	0.08	0.915
	As(V)	-	-	-	-	-
19AL	Cu	1667.41	318.40	0.07	0.05	0.947
	As(V)	1574.12	721.31	-	-	0.764
50AL	Cu	-	-	0.38	0.058	0.979
	As(V)	-	-	0.06	0.034	0.941
6S	Cu	-	-	0.63	0.12	0.946
	As(V)	1163.51	187.13	0.17	0.10	0.975
51S	Cu	-	-	0.49	0.028	0.988
	As(V)	1111.30	457.70	0.06	0.034	0.942
71S	Cu	272.31	174.11	0.50	0.03	0.999
	As(V)	-	-	0.10	0.06	0.944
Oak ash	Cu	-	-	-	-	-
	As(V)	-	-	-	-	-
Pine bark	Cu	-	-	0.08	0.02	0.970
	As(V)	1690.37	258.54	0.01	0.00	0.956
Mussel shell	Cu	-	-	-	-	-
	As(V)	1565.52	849.08	-	-	0.732

*bt*: Temkin isotherm constant; *Kt*: Temkin isotherm equilibrium binding constant; R^2^: coefficient of determination.

**Table 7 materials-15-05023-t007:** Values of Cu and As(V) adsorption (Q, in µmol kg^−1^) as well of of pH and DOC (in mg L^−1^) in the equilibrium solution for the various Cu and As(V) initial concentrations (C_0,_ in µmol L^−1^) added to the three by-products.

		Cu			As(V)		
Sorbent	C_0_	Q	pH	DOC	Q	pH	DOC
Oak ash	0.00	0.00	12.21	0.37	0.00	11.99	0.40
	100	3953.65	12.19	0.42	3945.07	12.00	0.40
	200	7619.05	12.22	0.45	7639.94	12.05	0.41
	400	15,455.80	12.18	0.36	14,451.22	12.09	0.82
	800	31,008.12	12.16	0.36	28,892.20	12.05	0.49
	1600	63,284.57	12.09	0.33	54,610.45	11.96	0.50
Pine bark	0.00	0.00	4.58	0.526	0.00	5.94	0.28
	100	3300.95	3.91	0.46	442.89	5.69	0.31
	200	6564.22	3.9	0.52	556.30	5.63	0.33
	400	13,112.66	3.82	0.43	1725.85	5.45	0.34
	800	24,405.27	3.77	0.44	3669.51	5.47	0.37
	1600	42,152.96	3.61	0.39	4036.64	5.17	0.30
Mussel shell	0.00	0.00	7.44	0.12	0.00	7.05	0.08
	100	3903.02	7.50	0.16	1003.36	7.63	0.24
	200	7797.85	7.53	0.18	2025.12	7.90	0.16
	400	15,088.63	7.50	0.18	4889.31	8.13	0.19
	800	29,335.28	7.42	0.19	7427.29	8.20	0.16
	1600	48,033.56	6.18	0.15	4515.32	8.33	0.11

## Data Availability

Not applicable.

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
