# Peer review of "Cu and As(V) Adsorption and Desorption on/from Different Soils and Bio-Adsorbents"

_materials, 2022, doi:10.3390/ma15145023_

Round 1

Reviewer 1 Report

This work reports on comparison of metal ion adsorption characteristics of several soils and bio adsorbents, by employing batch sorption and desorption experiments with Cu and As ions.

The study is performed on a high level, and especially the discussion of the results is rather detailed and informative.

The paper can certainly be recommended for publication in Materials.

Several points need to be be considered.

Major remarks.

1. The different sorption rates are discussed in correlating with the presence of various elements in the soils. However, rather surprisingly, the specific surfaces are not measured or presented.

Therefore, nitrogen sorption data and porosity characteristics should be supplied with the experimental data.

2. The sorption isotherms are fitted to various models, some of which fitted well, while others fitted poorly. It would be highly instructive to show the isotherm models in the non-linear plots, since visual observation allows the reader to grasp immediately the performance of the model equation.

3. Many of the isotherm graphs show that the concentration of the sorbent and/or the concentration range of the sorbate are chosen not optimally, which influences the accuracy of fitting of various models. If possible, additional experiments are needed to these cases, or otherwise, a discussion on the limited ranges chosen for these parameters.

Minor remark

the references need to be adjusted to the journal style.

Author Response

Reviewer 1:

Comments and Suggestions for Authors

This work reports on comparison of metal ion adsorption characteristics of several soils and bio adsorbents, by employing batch sorption and desorption experiments with Cu and As ions.

The study is performed on a high level, and especially the discussion of the results is rather detailed and informative.

The paper can certainly be recommended for publication in Materials.

RESPONSE: Thank you for your positive comments.

Several points need to be considered.

Major remarks.

  1. The different sorption rates are discussed in correlating with the presence of various elements in the soils. However, rather surprisingly, the specific surfaces are not measured or presented. Therefore, nitrogen sorption data and porosity characteristics should be supplied with the experimental data.

RESPONSE: Thank you for your comment. We have included data on surface area corresponding to the bio-adsorbents and soils in Supplementary Material.

  1. The sorption isotherms are fitted to various models, some of which fitted well, while others fitted poorly. It would be highly instructive to show the isotherm models in the non-linear plots, since visual observation allows the reader to grasp immediately the performance of the model equation.

RESPONSE: Thank you for your comment. Now we include Supplementary Material with new graphs, where the graphical fitting to the models was added.

  1. Many of the isotherm graphs show that the concentration of the sorbent and/or the concentration range of the sorbate are chosen not optimally, which influences the accuracy of fitting of various models. If possible, additional experiments are needed to these cases, or otherwise, a discussion on the limited ranges chosen for these parameters.

RESPONSE: Thank you for your comment. We have included further discussion and references in this regard (blue fonts).

Minor remark:

The references need to be adjusted to the journal style.

RESPONSE: Thank you for your comment. We have corrected it.

Reviewer 2 Report

The authors present an experimental study of Cu and As(V) adsorption and desorption into different soils and bio-adsorbents, particularly oak derived ash.

The article is well writen and an extensive discussion is performed for the mechanisms.

General comments:

English should be revised throughout the manuscript, i.e., line 286: the differences among the different

Please follow the journal’s reference template

The only possible weak point I could find is the originality of the research. Thus, the authors should add a thorough comparison of the results with previous research with other types of ash, such as https://doi.org/10.1007/s11356-013-2272-y

https://doi.org/10.1007/s11356-014-3032-3

 https://doi.org/10.1016/j.jhazmat.2020.122479 

https://doi.org/10.1016/j.scitotenv.2020.140205

Also, include in the introduction the ocurrence of several investigation where ash as been used as a succesful adsorbent of these types of ions.

Specific comments:

Line 113 and 117: “Elemente= exchangeable concentration; Elemento= concentration after extraction with ammonium” Is this right? Elemente elemento, please write in another way to be more clear for the reader.

Best regards

Author Response

Reviewer 2:

Comments and Suggestions for Authors

The authors present an experimental study of Cu and As(V) adsorption and desorption into different soils and bio-adsorbents, particularly oak derived ash. The article is well written and an extensive discussion is performed for the mechanisms.

RESPONSE: Thank you for your positive comments.

General comments:

English should be revised throughout the manuscript, i.e., line 286: the differences among the different

RESPONSE: Thank you for your comment. We have corrected all mistakes found (blue fonts).

Please follow the journal’s reference template

RESPONSE: Thank you for your comment. We have done it.

The only possible weak point I could find is the originality of the research. Thus, the authors should add a thorough comparison of the results with previous research with other types of ash, such as https://doi.org/10.1007/s11356-013-2272-y

https://doi.org/10.1007/s11356-014-3032-3

 https://doi.org/10.1016/j.jhazmat.2020.122479 

https://doi.org/10.1016/j.scitotenv.2020.140205

RESPONSE: Thank you for your comment. We have further discussed on these aspects as requested (blue fonts).

Also, include in the introduction the occurrence of several investigations where ash has been used as a successful adsorbent of these types of ions.

RESPONSE: Thank you for your comment. We have done it.

Specific comments:

Line 113 and 117: “Elemente= exchangeable concentration; Elemento= concentration after extraction with ammonium” Is this right? Elemente elemento, please write in another way to be more clear for the reader.

RESPONSE: Thank you for your comment. We have corrected it (blue fonts).

Reviewer 3 Report

Reviewer’s comments on the manuscript: “Cu and As(V) adsorption and desorption on/from different soils and bio-adsorbents” written by Raquel Cela-Dablanca et al.

The reviewed manuscript presents studies of the retention of Cu and As(V) in cultivated soils with different characteristics, as well as the capacity of different by-products (oak ash, pine bark, and mussel shell) to immobilize these contaminants. The obtained results of the research could be useful to program  appropriate practices to manage soils and low-cost by-products in order to reduce the risks of environmental contamination associated with the spreading of materials that contain both pollutants. In my opinion the manuscript is in the journal’s fields of interests. Moreover, it is interesting and well-written. Experiments are properly planned and the obtained data are clear. Presented discussion is also convincing. I recommend to accept it after minor revisions.

Things that should be improved/added before the publication:

·     All manuscript: It's probably the template's fault, but often words are not broken down correctly into syllables. Maybe it will be possible to correct it.

·       Abstract perfectly presents the relevance of the studies.

·      Introduction: I would like to encourage the Authors to add more specific information about the influence of As and Cu on humans health.

·       Some editorial mistakes, lines: 124, 339, 829.

·       Table 3, second line: L-1.

·       Tables: please present all numbers with the same number of significant digits.

·       Line 269: “All AL soils have a similar desorption of
both elements, while S-zone soils (with higher pH) desorb much more As than Cu (Fig. 4).” Could you add the explanation why is it so.

·       Maybe it is good idea to specify errors in Tables 5-7,  e.g.: error 1, error 2…

Author Response

Reviewer 3:

Comments and Suggestions for Authors

Reviewer’s comments on the manuscript: “Cu and As(V) adsorption and desorption on/from different soils and bio-adsorbents” written by Raquel Cela-Dablanca et al.

The reviewed manuscript presents studies of the retention of Cu and As(V) in cultivated soils with different characteristics, as well as the capacity of different by-products (oak ash, pine bark, and mussel shell) to immobilize these contaminants. The obtained results of the research could be useful to program appropriate practices to manage soils and low-cost by-products in order to reduce the risks of environmental contamination associated with the spreading of materials that contain both pollutants. In my opinion the manuscript is in the journal’s fields of interests. Moreover, it is interesting and well-written. Experiments are properly planned and the obtained data are clear. Presented discussion is also convincing. I recommend to accept it after minor revisions.

RESPONSE: Thank you for your positive comments.

Things that should be improved/added before the publication:

  • All manuscript: It's probably the template's fault, but often words are not broken down correctly into syllables. Maybe it will be possible to correct it.

RESPONSE: Thank you for your comment. It is the template. He hope it can be solved by the Journal.

  • Abstract perfectly presents the relevance of the studies.

RESPONSE: Thank you for your positive comment.

  • Introduction: I would like to encourage the Authors to add more specific information about the influence of As and Cu on humans health.

RESPONSE: Thank you for your comment. We have included further discussion in this regard (blue fonts).

  • Some editorial mistakes, lines: 124, 339, 829.
  • Table 3, second line: L-1.

RESPONSE: Thank you for your comment. Corrected (blue fonts)..

  • Tables: please present all numbers with the same number of significant digits.

RESPONSE: Thank you for your comment. Corrected.

  • Line 269: “All AL soils have a similar desorption of both elements, while S-zone soils (with higher pH) desorb much more As than Cu (Fig. 4).” Could you add the explanation why is it so.

RESPONSE: Thank you for your comment. As it is indicated in the 4.2 item of the manuscript: “For the soils of the present study, those with higher eCEC (S Soils) (Table 1) are those that desorbed less Cu (Fig. 4)”. Also to note that “By conducting a bivariate correlation study to find out how different soil parameters influence As(V) desorption, a significant (p <0.05) and positive correlation was obtained with soil pH (r = 0.837) and non-crystalline Fe and Al compounds (r = 0.848).” And finally: “As the pH increases, the positive charges on the colloidal surfaces (including those of the non-crystalline Fe and Al compounds) become negative, hindering the adsorption of As(V) in anionic form, causing the bonds to be more labile, while the opposite happens when acidity increases. This would explain the greater desorption of As(V) from S soils, with higher pH.

  • Maybe it is good idea to specify errors in Tables 5-7,  e.g.: error 1, error 2…

RESPONSE: Thank you for your comment. Corrected.

Round 2

Reviewer 1 Report

Authors totally misunderstand the recommendation to show the data and the non linear isotherm fits in new graphs. In the supplementary material they presented figures which look like Excel fits to polynomials, which have nothing common with the physical isotherm models of Equations 1,2,3. 

In the revised version, Authors included new citations to some of their previous papers, which apper to contain the same drawback: apparently Authors do not know how to plot such data, which may be the consequence of using of their statistical analysis software without graphical representation.

For example in their ref. 81.,  Khezami, L.; Capart, R, such isotherms are shown.  Same can be found in many many papers dealing with metal ion adsorption.

I strongly insist to present the fits of the non-linear isotherms in the main text, and not in the suppl material, for example adding the theoretical curves to the experimental data in figures 1 and 2.
